

# Pest control of aphids depends on landscape complexity and natural enemy interactions

Emily A. Martin[1], Björn Reineking[2,3,4], Bumsuk Seo[4,5] and Ingolf Steffan-Dewenter[1]

[1] Department of Animal Ecology and Tropical Biology, Biocenter, University of Würzburg, Am Hubland, Würzburg, Germany
[2] Irstea, UR EMGR, St-Martin-d'Hères, France
[3] Université Grenoble Alpes, Grenoble, France
[4] Biogeographical Modelling, Bayreuth Center of Ecology and Environmental Research BayCEER, University of Bayreuth, Bayreuth, Germany
[5] Department of Plant Ecology, University of Bayreuth, Bayreuth, Germany

Corresponding author
Emily A. Martin,
emily.martin@uni-wuerzburg.de

## ABSTRACT

Aphids are a major concern in agricultural crops worldwide, and control by natural enemies is an essential component of the ecological intensification of agriculture. Although the complexity of agricultural landscapes is known to influence natural enemies of pests, few studies have measured the degree of pest control by different enemy guilds across gradients in landscape complexity. Here, we use multiple natural-enemy exclosures replicated in 18 fields across a gradient in landscape complexity to investigate (1) the strength of natural pest control across landscapes, measured as the difference between pest pressure in the presence and in the absence of natural enemies; (2) the differential contributions of natural enemy guilds to pest control, and the nature of their interactions across landscapes. We show that natural pest control of aphids increased up to six-fold from simple to complex landscapes. In the absence of pest control, aphid population growth was higher in complex than simple landscapes, but was reduced by natural enemies to similar growth rates across all landscapes. The effects of enemy guilds were landscape-dependent. Particularly in complex landscapes, total pest control was supplied by the combined contribution of flying insects and ground-dwellers. Birds had little overall impact on aphid control. Despite evidence for intraguild predation of flying insects by ground-dwellers and birds, the overall effect of enemy guilds on aphid control was complementary. Understanding pest control services at large spatial scales is critical to increase the success of ecological intensification schemes. Our results suggest that, where aphids are the main pest of concern, interactions between natural enemies are largely complementary and lead to a strongly positive effect of landscape complexity on pest control. Increasing the availability of seminatural habitats in agricultural landscapes may thus benefit not only natural enemies, but also the effectiveness of aphid natural pest control.

## INTRODUCTION

Pest control by natural enemies is an essential ecosystem service valued at $4.49 billion annually in the USA alone (*Losey & Vaughan, 2006*). In the context of increasing pressure to reduce harmful chemical inputs while maintaining or increasing current agricultural yields, the gradual replacement of conventional agricultural practices with natural pest control provided by functional ecosystems is a major hope for the future (*Bommarco, Kleijn & Potts, 2013*). However, current understanding of the flow and stability of pest control services in human-modified systems is limited, and constrains our ability to implement an ecological intensification of agriculture (*Rusch et al., 2013*).

The distribution and abundance of agricultural pests and their natural enemies are known to be influenced both by local management intensity (*Bengtsson, Ahnström & Weibull, 2005*) and by the landscape context surrounding crop fields (*Bianchi, Booij & Tscharntke, 2006*; *Rand, Van Veen & Tscharntke, 2012*). In landscapes with high amounts of complex or seminatural habitats, enemies such as parasitoid wasps, predatory beetles, and insectivorous birds are frequently more abundant and species-rich than in simplified landscapes with large monocultures and little complex habitat. These effects vary according to the enemies considered and the spatial scale of observations (*Chaplin-Kramer et al., 2011*), and are strongest for enemies that depend on permanent habitat structures for nesting, overwintering and other food resources (*Bianchi, Booij & Tscharntke, 2006*). Within crop fields, these landscape effects can be intensified by spillover, defined as the flow of organisms occurring across the interface between crop and non-crop habitats (*Tscharntke, Rand & Bianchi, 2005*). In contrast, effects of landscape context on pest distributions are less well understood and appear highly variable between systems and years (*Bianchi, Booij & Tscharntke, 2006*; *Chaplin-Kramer et al., 2011*; *O'Rourke, Rienzo-Stack & Power, 2011*; but see *Meehan et al., 2011*). As pest distributions are a reflection of landscape effects on the pests themselves, but also of their suppression by natural enemies in the same landscape over more than one season (*O'Rourke, Rienzo-Stack & Power, 2011*; *Chaplin-Kramer & Kremen, 2012*), estimating the effects of landscape context on pests and thus on pest suppression requires distinguishing pest abundance in the presence, vs. in the absence of enemies. However, the direct effect of landscape context on the strength of pest suppression by natural enemies has only rarely been measured (*Chaplin-Kramer et al., 2011*) and may not show a direct relationship to the abundance and richness of natural enemies in the same landscapes (*Letourneau et al., 2009*).

Effects of natural enemies on pests can involve multiple interactions that prevent them from being deducible from the diversity of the natural enemy community (*Rand, Van Veen & Tscharntke, 2012*; *Martin et al., 2013*). These interactions can be additive or synergistic, i.e., pest control by multiple enemies is as strong or stronger than the sum of each enemy's effect in isolation; neutral, i.e., pest control is similar whether enemies are combined or not; or antagonistic, i.e., negative interactions such as intraguild predation or behavioral interference lead to lower pest control when enemies are combined than in isolation (*Letourneau et al., 2009*). These effects are determined by a variety of possible mechanisms, including niche complementarity and functional redundancy in addition to

intraguild predation (*Straub, Finke & Snyder, 2008*; *Tylianakis & Romo, 2010*), and may also be affected by the complexity of local vegetation structures (*Janssen et al., 2007*). However, in terrestrial systems, interactions of predator species and functional groups have almost exclusively been investigated at small spatial and temporal scales that do not reflect the complexity of real-world landscapes (*Duffy et al., 2007*; but see *Thies et al., 2011*; *Holland et al., 2012*). Results of these local studies show highly unpredictable interactions across systems and organisms, and little consensus has been reached on the factors that determine interaction strength and direction in real-world ecosystems (*Tylianakis & Romo, 2010*). Thus, although pest control requires the presence of natural enemies in the system, its effectiveness can only be approximated by simple measures of enemy community composition (abundance, species richness) if the direction of enemy interactions is known across all relevant spatial and temporal scales. Quantifying effective pest control across landscapes thus requires measuring the effects of the entire pool of enemies; predicting variations of these effects requires identifying the individual contributions of different enemies to pest control, and the nature of their interactions across landscapes (*Martin et al., 2013*). For this, definition of complexes of natural enemies (guilds) of similar body size, mobility and foraging habits is a meaningful and, importantly, realistically applicable proxy (*Macfadyen, Davies & Zalucki, 2014*).

Aphids are a major pest of crops worldwide. Several groups of natural enemies are known to provide control of aphid pests (*Schmidt et al., 2003*). They include parasitoid wasps and syrphid fly larvae ("flying insects"), which colonize aphid-infested plants by flying onto them as adults and ovipositing in or near aphids. More generalist enemies include carabid and staphylinid beetles and spiders, which mainly colonize plants from the ground and occupy a lower stratum than flying insects ("ground-dwellers"; *Schmidt et al., 2003*). In addition, birds represent the top predators for insects in many agricultural systems. Direct effects of bird predation on aphids have rarely been assessed (but see *Tremblay, Mineau & Stewart, 2001*), but their potential role as antagonists of other enemies may seriously impact overall amounts of pest control (*Martin et al., 2013*).

In this study, we use multiple natural enemy exclosures on pests of cabbage *Brassica oleracea* var. *capitata* in a replicated design across landscapes of an agricultural region in South Korea. We examined the effects of three main functional guilds of natural enemies on pest control of aphids across a gradient in landscape complexity. The three enemy functional guilds distinguished here are birds and other vertebrates larger than 1.5 cm; flying insects including syrphid flies, parasitoid and predatory wasps; and ground-dwellers including spiders, carabid and staphylinid beetles. Specifically, we aimed (1) to assess the strength of pest control by all natural enemies combined on aphid populations, across a gradient in landscape complexity; (2) to identify the separate contributions of functional guilds of natural enemies to pest control at the landscape scale and the nature of enemy interactions across landscapes and (3) to evaluate the importance of local management intensity for these effects. We hypothesized that interactions of the natural enemy pool may change with landscape complexity due to changing amounts of seminatural habitat as a refuge against intraguild predation (*Janssen et al., 2007*), contiguity of more distinct

habitats intensifying spillover, and changing density ratios between pests, enemies, and among enemy functional guilds (*Bianchi, Booij & Tscharntke, 2006*; *Chaplin-Kramer et al., 2011*); and that this has consequences for overall pest control strength (*Martin et al., 2013*).

## MATERIALS AND METHODS

**Study sites and landscapes.** This experiment was performed from July to September 2010 in the 55 km$^2$ agricultural region of Haean, South Korea (long. 128°5′ to 128°11′E, lat. 38°13′ to 38°20′N; Fig. S1). This region is located at the head of the Soyang Lake watershed, a major water and energy source for the northern half of South Korea. In this region, annual and perennial crops are cultivated in fields <1 ha separated by seminatural margins. Patches of deciduous forest and riparian corridors are distributed throughout, contributing to high landscape heterogeneity.

Eighteen fields were selected in this region (*Martin et al., 2013*), including 16 fields inside the Haean catchment (Fig. S1) and two fields 20 km to the south, in an area with similar land use and climatic conditions. Of these, 13 were managed organically and five conventionally. Due to the difficulty of convincing farmers to perform experiments in their fields, the number of organic vs. conventional fields could not be balanced in this design. Nine fields were planted with a Brassicaceae crop and nine with one of five other crop families. Planting dates of these crops varied from 0 to 55 days (22 ± 20, mean ± SD) before the start of the experiment. Fields were distant by a minimum of 211 m (distance from field edge), with a mean inter-field distance of 3.2 ± 0.1 km (mean ± SE). The field locations covered a gradient in landscape complexity from 6.3% to 43.3% seminatural habitat in a 700 m radius around fields (see below), and bordered on seminatural margins of similar size and plant composition. Seminatural habitat was defined in this region as including seminatural field margins, secondary regrowth and shrubs, 1 and 2-year old fallows, and forest edges (2 m-wide buffers). Its proportion around fields was calculated using polygon maps of the region (ground-truthed in 2009 and 2010; *Seo et al., 2014*) in ArcGIS 9.3 and R Statistical Software v. 2.13.1 (*R Core Team, 2013*).

**Experiment.** Within each field, one 20 m$^2$ plot was reserved for the experiment and left uncropped. It was separated from the surrounding crop by a plastic barrier and planted with cabbage *Brassica oleracea* var. *capitata*. No pesticides were applied on these plots. After an initial 20 days, six rows of four cabbages were randomly marked in each plot from which all herbivores were removed. Six natural enemy exclusion treatments were installed on these plants between day 20 and 21 and maintained until harvest after 60 days.

We initialized the experiment by inoculating aphids in these treatments. In order to enable comparison between the six treatments of a given plot, the same starting number of aphids was inoculated in all the treatments of one plot (*Chaplin-Kramer & Kremen, 2012*). This was done by placing on each treated cabbage the average number of aphids per plant found in the plot. As densities were low at the start of the season, aphids thus inoculated in each plot varied from 2 to 10 per plant. In plots where no aphids were present at the beginning of the experiment, a minimum of two adult aphids per plant was inoculated. This procedure was selected in preference to inoculation of the same number of aphids

in all plots, in order to increase representativeness in the experiment of the pest pressure occurring in each plot (*Thies et al., 2011*). Cabbage leaves infested with laboratory-reared cabbage aphids *Brevicoryne brassicae* (Linné) were transported to the field and fragments with the approximate number of aphids were deposited on treated cabbages. After one day, successful transfer of the desired number of aphids to treated plants was verified and any aphids in excess were removed. Initial aphid densities were unrelated to either the percent seminatural habitat around fields (Poisson linear model corrected for overdispersion, $n = 18$, $t = 0.3$, $p = 0.8$), the crop type of surrounding fields ($t = 1.2$, $p = 0.2$) or their management ($t = -0.3$, $p = 0.8$). Initial densities tended to be positively related to the maturity of the surrounding crop, but the relationship was not significant ($t = 1.9$, $p = 0.08$).

Starting 10 days after initializing the treatments, arthropods were monitored at three occasions (10 day intervals) throughout the growth period, by carefully inspecting both sides of leaves and recording the number, species and life stage of arthropod herbivores, parasitoids (parasitized aphid mummies) and predators, mainly the larvae of syrphid flies. Sap-sucking species included cabbage aphids *Brevicoryne brassicae*, green peach aphids *Myzus persicae* (Sulzer) and low densities of the turnip aphid *Lipaphis erysimi* (Kaltenbach). Larvae of the leaf-chewing Lepidoptera *Pieris rapae*, *Pieris brassicae* (Linné) and *Trichoplusia ni* (Hübner) were also present and their effects are described in a separate publication (*Martin et al., 2013*); see below. After 60 days, cabbage plants were harvested and weighed for fresh biomass. As one plot was monitored on two occasions only and monitoring data from one plant are missing, the total number of observations is 17 plots × 6 treatments × 4 plants × 3 sampling dates +1 plot × 6 treatments × 4 plants × 2 sampling dates −1 plant = 1,271. Where *Martin et al. (2013)* present results for Lepidopteran pests only, the present study focuses on the response of aphid pests. The final biomass of cabbages is the only measure common to both studies, and is examined here in relation to aphids and their enemies.

**Field exclusion treatments.** Natural enemy exclusion treatments were cages designed to exclude combinations of three guilds of natural enemies: G—ground-dwellers (spiders, carabid and staphylinid beetles), F—flying insects (syrphid flies, parasitoid and predatory wasps), and B—birds and other vertebrates larger than 1.5 cm. Cages were 50 ∗ 150 ∗ 100 cm and covered one row of four cabbage plants. We used combinations of chicken wire (1.5 cm mesh size), fine polyester mesh (0.8 mm) and plastic barriers coated with insect glue to exclude either all enemies (treatment -G-F-B, "no enemy" control), birds and flying insects (-F-B), ground-dwellers and birds (-G-B), only ground-dwellers (-G), only birds (-B), or no enemies (O, open control). Cage treatments and corresponding symbols are summarized in Fig. S2. Although designed to exclude or allow access to specific enemy guilds with relevance for pest control, these treatments do not allow to distinguish the effect of increasing enemy diversity *per se*, vs. increasing enemy density due to access by a higher number of guilds (additive design *sensu* *Tylianakis & Romo, 2010*). However, they do provide insight on the relative importance of separate guilds, and on the interactions taking place between guilds in terms of their outcome for pest control (*Schmidt et al., 2003*).
An additional treatment excluding both enemies and herbivores controlled for abiotic soil conditions between plots; ecofriendly pesticide was applied at the start of the experiment in this treatment only. Differences in soil conditions between plots had no effect on final cabbage biomass (*Martin et al., 2013*). Although fine mesh cages were effective at excluding most natural enemies including parasitoids, they were not impermeable to external colonization by aphids, as indicated by test cages without inoculation (E Martin, pers. obs., 2010). At the start of the experiment, two live pitfall traps were installed in all treatments excluding ground-dwellers. After initial collection of the ground arthropods already present, pitfall traps remained empty throughout the experiment. This method was effective also for spider exclusion (*Martin et al., 2013*). Microclimatic and light differences between treatments were tested by comparing values inside and outside fine mesh treatments in each plot (*Martin et al., 2013*). Light transparency of fine mesh treatments was $83 \pm 0.9\%$ (mean $\pm$ s.e.m). Air humidity did not differ significantly between the inside and outside of cages ($t = 1.4, p = 0.3$ and $t = 0.04, p = 0.7$ on sunny and rainy days, respectively). Temperatures were $0.56 \pm 0.1\,°C$ higher inside than outside fine mesh treatments (mean $\pm$ s.e.m; $t = 5.5, df = 17, p < 0.001$). However, microclimatic and light differences had no significant effect on plant growth between treatments (*Martin et al., 2013*). In addition, any error caused by these differences would have occurred in all plots, and thus should not affect result interpretation at the landscape scale. As only 2.2% of aphids were winged (700 out of 31,503 counted individuals), the role of aphid dispersal appears to have been negligible (*Thies et al., 2011*).

**Data analysis.** Aphid population growth, parasitism rate, syrphid fraction and final crop biomass were analyzed using linear and generalized linear mixed models in R Statistical Software 2.13.1 (*R Core Team, 2013*).

Average daily aphid population growth ($n = 1,271$ data points; Methods §2) was calculated as the $\log(N + 1)$-ratio per day of aphid densities (sampling dates 1 to 3) to initial densities. For example, at sampling date 1, aphid population growth was $[\log(N_{\text{aphids at date 1}} + 1) - \log(N_{\text{aphids initial}} + 1)]/$(number of days from the start of the experiment to date $1 = 10$). At date 2, it was $[\log(N_{\text{aphids at date 2}} + 1) - \log(N_{\text{aphids initial}}+1)]/$(number of days from the start of the experiment to date $2 = 20$). These measures of population growth thus reflect either relatively short-term (until date 1, 10 days), mid-term (until date 2, 20 days) or long-term (until date 3, 30 days) changes in aphid populations over time.

Here, as in other studies quantifying aphid pest control on a landscape gradient (e.g., *Holland et al., 2012*; *Rusch et al., 2013*; *Thies et al., 2011*; but see *Chaplin-Kramer & Kremen, 2012*), the calculation of aphid density and population growth accounts in practice for both processes of growth and colonization from the surrounding area. In this study, colonization occurred in all treatments (see Methods §3). Although differences in colonization between treatments could not be quantified, any hindrance of colonization by fine mesh cages would lead to an underestimation, not an overestimation, of actual pest control. Indeed, if treatments accessible to enemies were more colonized than fine mesh controls, then enemies appear to have reduced aphids less than they "truly"

have (final densities after predation = remaining aphids after predation + additional colonized aphids). Here, aphid population growth (defined as the outcome of growth and colonization by all aphid species) was used as a response variable instead of aphid density, in order to account for a variable number of aphids initially present in each plot (*Rusch et al., 2013*; *Thies et al., 2011*).

Aphid population growth was modelled using a linear mixed model in package nlme (*Pinheiro et al., 2013*) and variance functions were included to model heteroscedasticity. Parasitism rates (the ratio of parasitized to all aphids) and syrphid fractions (the ratio of syrphids to total aphids + syrphids) were modelled using a binomial response with logit link in package lme4 (*Bates, Maechler & Bolker, 2013*) . Observation-level random effects were included to account for overdispersion (final overdispersion parameter $\Psi < 0.2$). Crop biomass ($n = 432$ plants) was modelled using a Gamma error with log link in package lme4. All models included 'exclusion treatment' (6 levels of natural enemy exclusion) nested within 'plot' (18 plots, each in one landscape sector) as random effects, in order to account for pseudoreplication within each plot and exclusion treatment. Thus, all plants within the same exclusion treatment are treated as non-independent replicates of that treatment. For all responses except biomass (see below), explanatory variables included exclusion treatment, percent seminatural habitat in a radius around each plot, sampling date (1–3), management type of the surrounding field (organic/conventional), crop type (Brassicaceae/non Brassicaceae) and crop maturity of the surrounding field, and 2-way and 3-way interactions. Management, crop type and crop maturity did not correlate significantly either with percent seminatural habitat or with each other (Pearson's $r$ always $<0.3, p > 0.2$).

Aphid population growth and pest control may have been influenced not only by environmental variables, but also by local aphid densities occurring in each plot. In order to identify density-dependent effects of the number of aphids present, a second set of models was constructed with the additional explanatory variable 'initial number of aphids' and interactions with 'sampling date' and 'treatment'. The initial number of aphids for a given sampling date was the number of aphids counted at the previous date (dates 2 and 3), or the number of aphids initially inoculated (date 1 and for average daily aphid population growth). The initial number of aphids was selected in 95% model confidence sets (see below) with a probability of 1 for aphid population growth, 0.93 for parasitism and 0.29 for syrphid fractions (Table S1). Effects of the initial number of aphids on aphid growth were negative for all sampling dates (averaged $b = -0.006 \pm 0.002$). Effects on parasitism were positive in treatments accessible to parasitoids ($b = 0.46 \pm 0.42$ in -G-B), and slightly positive on syrphid fractions ($b = 0.003 \pm 0.055$). Results of other explanatory variables and their interpretation were not affected by inclusion of this factor, thus the initial number of aphids was not included in subsequent analyses (Table S1).

In order to determine the most adequate spatial scales for analysis of each response variable, Akaike's Information Criterion with a correction for finite sample sizes (AICc) was used to compare the final models at 100 m-intervals between 100 m and 1,000 m around fields. Lowest AICc values were selected at 700 m for aphid population growth,

200 m for parasitism rates and 900 m for syrphid fractions. Results are shown for these most predictive scales. However, effects of landscape and landscape:treatment interactions were also selected with similar effects in model confidence sets (see below), at all but three other scales for aphid population growth (Table S2).

Model selection was performed by assembling a 95% confidence set of models (cumulated sum of AICc weights ≤95%) from the set of all possible models, using sequential AICc testing with the function "dredge" in R package MuMIn (*Barton, 2012*). Model averaging was performed on this set and weights were calculated for each explanatory variable as the sum of the AICc weights of each model it occurs in. Weights of each explanatory variable can thus be interpreted as the probability of its presence, or importance, within the global averaged model (*Burnham, Anderson & Huyvaert, 2011*). Model-averaged coefficients of explanatory variables were used to plot the predicted values of responses. Tukey multiple comparisons of means were performed on models without interactions, and slope comparisons of models with interactions were performed using manually defined contrast matrices. *P*-values of multiple comparisons were adjusted for the False Discovery Rate (*Benjamini & Yekutieli, 2001*). Models were checked graphically for violation of assumptions of normality and homoscedasticity. Spline correlograms of Moran's I against distance confirmed that any spatial autocorrelation present in the raw data (for instance due to overlapping landscape sectors) was accounted for by inclusion of the model random effects (*Zuur et al., 2009*), thus potential non-independence of sites due to spatial proximity was accounted for in models. Further, temporal autocorrelation due to non-independence of sampling dates was addressed by including a correlation structure in models for aphid population growth of the form corAR1 (∼Date | Plot_ID/Treatment_ID/Plant). In generalized mixed models in lme4, such structures are not yet implemented. For parasitism rates and syrphid fractions, we thus accounted for effects of repeated measures by including 'sampling date' as an additional random effect nested in 'exclusion treatment' and 'plot'. Results for aphid population growth and syrphid fractions were unchanged by inclusion of these structures. For parasitism rates, they led to a difference in the scale of response (200 m instead of 1,000 m) but to no change in result interpretation. Thus, effects were considered robust for all response variables to possible effects due to non-independence of sampling dates.

Ultimately, the final measure of pest control for farmers is the biomass of the crop. In this experiment, final crop biomass decreased with increasing landscape complexity in all treatments except -G-B, reflecting the impact of stronger herbivory by Lepidopteran pests in complex landscapes, as previously shown elsewhere (Pearson's $r = -0.53$, $P < 0.001$ between biomass and herbivory by Lepidoptera; *Martin et al., 2013*). In order to estimate, in turn, the importance of aphid suppression for final biomass provision, we used a separate set of models relating aphid population growth and final crop biomass, respectively, to syrphid fractions and parasitism rates. Tested effects of aphid population growth on biomass were not modified by inclusion of Lepidopteran herbivory in models.

Cited values of mean reduction in aphid population growth across sampling dates compared to controls without enemies were calculated from model predicted values as $(R_{\text{treatment}}\text{-}R_{\text{control}})/R_{\text{control}}$ where $R =$ mean aphid population growth in all landscapes.

## RESULTS

### Effects of enemy guilds and landscape context on aphid pest control

On the first sampling date, aphid densities averaged $44.3 \pm 6.2$ individuals/plant (minimum 0, maximum 1,105; $n = 1,271$ plants in 18 plots). Mean densities decreased in the following dates to $20.6 \pm 4.4$ (min. 0, max. 1093) and $8.6 \pm 2.3$ (min. 0, max. 503) aphids/plant on dates 2 and 3, respectively. These were reflected by average population growth rates, which were positive in some treatments after 10 days (sampling date 1) but negative or zero over the whole season (30 days, sampling date 3; Fig. 1).

On average, daily aphid population growth was four times (range 1–6 times) higher in the absence of all natural enemies than in their presence (Fig. 1 and Table 1). This effect was maintained across sampling dates, with overall population growth lowest in treatments accessible to natural enemies until the end of the season. Individual enemy guilds reduced aphid population growth to lower values than in their absence: on average, growth rates were 2.3 times (0.2–4) and 3.3 times (0.9–5) higher in the absence of ground-dwellers and flying insect enemies respectively, than in their presence individually. Direct effects of vertebrate predators (birds) are not measurable in isolation from flying insect enemies. However, the exclusion of birds from treatments with other enemy guilds did not significantly impact mean aphid population growth at any date (O vs. -B and -G vs. -G-B; Fig. 1).

The combined effects of ground-dwellers and flying insects on aphid suppression were stronger than in isolation (Fig. 1). This result was particularly present in the first phase of the experiment (until date 1), and lessened over time (dates 2 & 3). On average, population growth was 1.9 times (1.4–2.6) higher with ground-dwellers alone than in the presence of both guilds, and 0.8 times (0.1–1.7) higher with flying insects alone than in the presence of both guilds. Overall, ground-dwellers and flying insects thus had complementary impacts on aphid suppression.

Aphid population growth in controls excluding all natural enemies increased from simple to complex landscapes (Fig. 1, Fig. S3 and Table S3: -G-F-B vs. zero, $p_{\text{adjusted}} < 0.05$ for sampling dates 1 & 2). However, the degree of aphid suppression by natural enemies also increased with landscape complexity (landscape:treatment interaction; Table 1 and Table S3), and these effects were maintained across several spatial scales (Table S2). At the 700 m scale around fields, aphid suppression by all natural enemies (the difference between population growth in the absence and in the presence of all enemies) was ca. six times higher in complex than in simple landscapes (mean $\pm$ s.e.m. across sampling dates $5.6 \pm 2.5$, from a landscape with 18% to a landscape with 45% seminatural habitat; Table S3: the slope of O is significantly lower than the slope of -G-F-B at dates 1 and 2, thus the difference between O and -G-F-B increases with landscape complexity). This effect was maintained in treatments combining multiple guilds (-G, -B). Further, effects

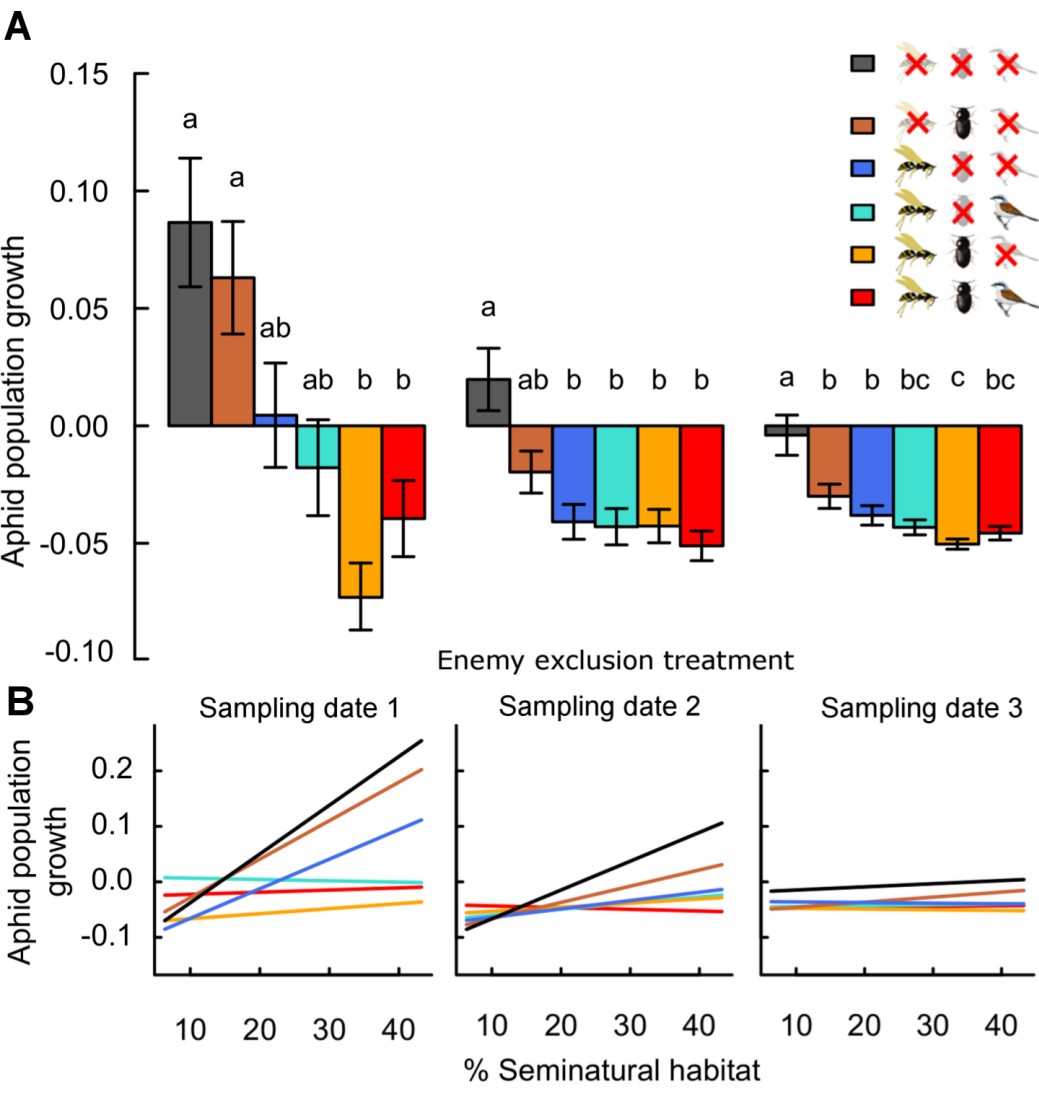

**Figure 1** **Effects of landscape complexity and natural enemy exclusion on average daily aphid population growth across three sampling dates (1–3).** Average daily population growth is the log-ratio of aphid density on sampling dates 1 to 3 and of initial aphid density, divided by the number of days since the start of the experiment (date 1: after 10 days, date 2: after 20 days, date 3: after 30 days). Part A shows mean growth ± s.e.m. per exclusion treatment and sampling date. Part B shows variation of predicted values across the gradient in landscape complexity, measured by percent seminatural habitat in a 700 m radius around fields. Data points per treatment and date are provided in Fig. S3. See Table S3 for multiple slope comparisons. Different letters indicate significant differences between treatments. In the legend, crossed-out symbols indicate exclusion of natural enemy functional guilds. Treatments remain accessible to non-excluded guilds. Guilds of natural enemies include flying insects (parasitoids, syrphid flies and predatory wasps; wasp symbol), ground-dwellers (carabid beetles, staphylinids and spiders; beetle symbol) and birds (and other vertebrates larger than 1.5 cm; bird symbol).

of ground-dwellers only (the difference between treatments with ground-dwellers only and controls without enemies) were less strong across the landscape gradient than effects of flying insects (Table S3: -G-F-B vs. -F-B and vs. -G-B). In complex landscapes, flying insect enemies reduced aphid growth ca. two times more than ground-dwelling predators

Martin et al. (2015), *PeerJ*, DOI 10.7717/peerj.1095

**Table 1 Results of model selection relating landscape complexity and enemy exclusion to response variables.** Model lists show the 95% best models at the most predictive scale for each response variable. The sum of weights for each term is the sum of AIC weights of all models selecting it and represents the probability of being present in the 95% model confidence set.

| Response | no | Model specification | | | | | | | | | df | AICc | Δ AIC | w | w 95% |
|---|---|---|---|---|---|---|---|---|---|---|---|---|---|---|---|
| Aphid population growth ~ | M1 | M + | L + | D + | T + | M:D + | L:D + | T:D + | L:T + | L:T:D | 52 | −3,480 | 0 | 0.94 | 1 |
| *Sum of weights* | | *1* | *1* | *1* | *1* | *1* | *1* | *1* | *1* | *1* | | | | 0.94 | 1 |
| Parasitism rate ~ | M2 | M + | L + | D + | T + | M:D + | L:D + | | L:T | | 23 | 1486.8 | 0.00 | 0.71 | 0.75 |
| | M3 | M + | L + | D + | T + | M:D + | | | L:T | | 21 | 1491.6 | 4.87 | 0.06 | 0.07 |
| | M4 | | L + | D + | T + | | L:D + | | L:T | | 20 | 1491.8 | 5.03 | 0.06 | 0.06 |
| | M5 | M + | L + | D + | T + | | L:D + | | L:T | | 21 | 1492.1 | 5.34 | 0.05 | 0.05 |
| | M6 | M + | L + | D + | T + | M:D + | L:D + | T:D + | L:T | | 33 | 1492.8 | 6.03 | 0.04 | 0.04 |
| | M7 | M + | L + | D + | T + | M:D + | L:D | | | | 18 | 1493 | 6.26 | 0.03 | 0.03 |
| *Sum of weights* | | *0.94* | *1* | *1* | *1* | *0.89* | *0.14* | *0.04* | *0.97* | | | | | 0.95 | 1.00 |
| Syrphid fraction ~ | M8 | M + | L + | D + | T + | | | | L:T | | 19 | 1229.6 | 0.00 | 0.23 | 0.25 |
| | M9 | | L + | D + | T + | | | | L:T | | 18 | 1,230 | 0.35 | 0.20 | 0.21 |
| | M10 | M + | L + | D + | T + | M:D + | | | L:T | | 21 | 1230.2 | 0.58 | 0.18 | 0.19 |
| | M11 | M + | L + | D + | T + | M:D + | L:D + | | L:T | | 23 | 1232.1 | 2.47 | 0.07 | 0.07 |
| | M12 | M + | | D + | T + | M:D | | | | | 15 | 1232.4 | 2.78 | 0.06 | 0.06 |
| | M13 | M + | | D + | T + | | | | | | 13 | 1232.5 | 2.82 | 0.06 | 0.06 |
| | M14 | M + | L + | D + | T + | | L:D + | | L:T | | 21 | 1233 | 3.36 | 0.04 | 0.05 |
| | M15 | | L + | D + | T + | | L:D + | | L:T | | 20 | 1233.2 | 3.58 | 0.04 | 0.04 |
| | M16 | | | D + | T + | | | | | | 12 | 1233.3 | 3.65 | 0.04 | 0.04 |
| | M17 | M + | L + | D + | T + | M:D | | | | | 16 | 1234.3 | 4.67 | 0.02 | 0.02 |
| *Sum of weights* | | *0.71* | *0.84* | *1* | *1* | *0.35* | *0.16* | | *0.81* | | | | | 0.93 | 1.00 |

**Notes.**

Selected explanatory variables are M, management type of the nearest surrounding field (organic/conventional); L, landscape complexity (% seminatural habitat in the surrounding radius); D, sampling date (1–3); T, Exclusion treatment (6 levels of natural enemy exclusion); w, AIC weight compared to all possible models; w95%, AIC weight within the 95% model confidence set.

(mean ± s.e.m. across sampling dates 2.1 ± 0.4; reduction compared to controls for flying insects and ground-dwellers, respectively).

## Parasitism rates and syrphid fractions

Rates of parasitism and syrphid fractions were higher in treatments accessible to flying insects than in treatments excluding them, confirming the effectiveness of exclosures for these enemies (Fig. 2 and Table 1). These differences were significant for parasitism rates and less strong for syrphid fractions, as only low numbers of syrphid larvae (on average 0.3 ± 0.03 in accessible treatments) were recorded per plant. Rates of parasitism and syrphid fractions increased with landscape complexity mainly in treatments accessible to flying insects only (Fig. 2; Table S3: -G-B vs. zero). Although reduction of aphid population growth was strongest in complex landscapes when flying insects were combined with other guilds (Fig. 1), this was not reflected by similar high parasitism or syrphid fractions in complex landscapes in treatments combining several guilds (Fig. 2). This difference thus suggests a negative effect of bird and ground-dweller access on the effectiveness of flying insect enemies.

## Aphid population growth and yields

Aphid population growth was strongly negatively correlated with parasitism rate and syrphid fractions (Fig. 3), confirming the impact of these enemies for reduction of aphid populations. Neither population growth nor cumulated aphid densities led to a significant decrease in final crop biomass. However, high syrphid fractions tended to have a positive impact on biomass, thus indicating a link via pest suppression between the proportion of syrphid predators and the provision of yields (Fig. 3). This is reinforced by a significantly positive link between biomass and syrphid densities ($Chi^2 = 4.5$, $p = 0.03$, $n = 431$). In contrast, neither parasitism rate nor parasitoid density significantly affected crop biomass ($Chi^2 = 0.6$, $p = 0.4$ and $Chi^2 = 2$, $p = 0.2$, respectively).

## Management effects

After the start of the experiment, neither the crop type of surrounding fields (Brassicaceae vs. non Brassicaceae) nor their maturity had an impact on aphid population growth or enemy rates (Table 1). However, the management intensity of surrounding fields (organic vs. conventional) affected response variables. Particularly at sampling date 1, population growth was higher in plots surrounded by conventional than by organic fields (Fig. S3; Table 1). However, no differences were found at subsequent dates (Fig. S3). Aphid populations thus decreased more strongly from date 1 to date 3 in fields surrounded by conventional than by organic fields. In contrast, enemy densities and parasitism rates were similar in both management types at date 1, but higher near conventional fields on the following dates (Figs. S4 and S5).

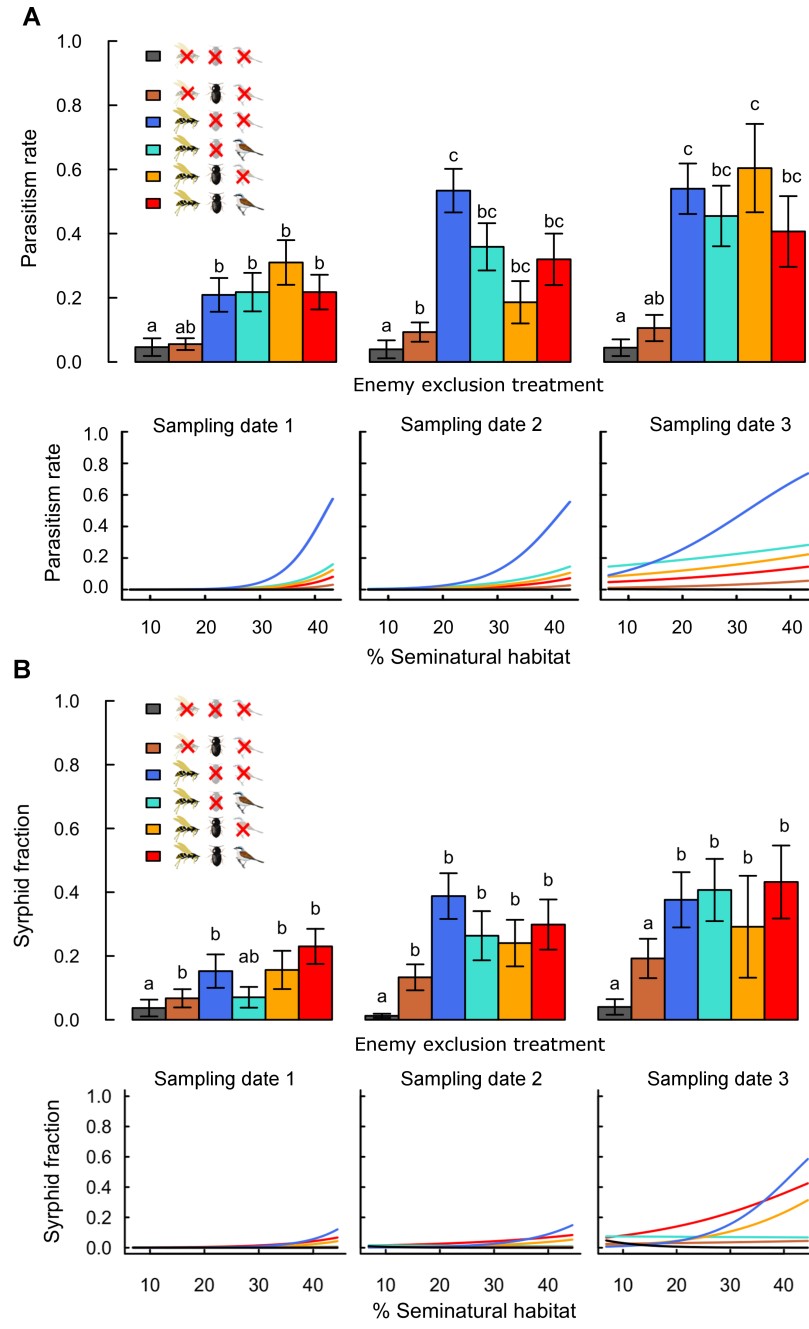

**Figure 2 Effects of landscape complexity and natural enemy exclusion on (A) aphid parasitism rate and (B) syrphid fraction at three sampling dates (dates 1–3; 10 day intervals).** The top half of each figure shows mean values ± s.e.m. per exclusion treatment and sampling date. The lower half shows variation of predicted values across the gradient in landscape complexity, measured by percent seminatural habitat in a 200 m and 900 m radius around fields for parasitism rates and syrphid fractions, respectively. Data points per treatment and date are provided in Figs. S4 and S5. See Table S3 for slope multiple comparisons. Different letters indicate significant differences between treatments. Detailed legend description is provided in Fig. 1.

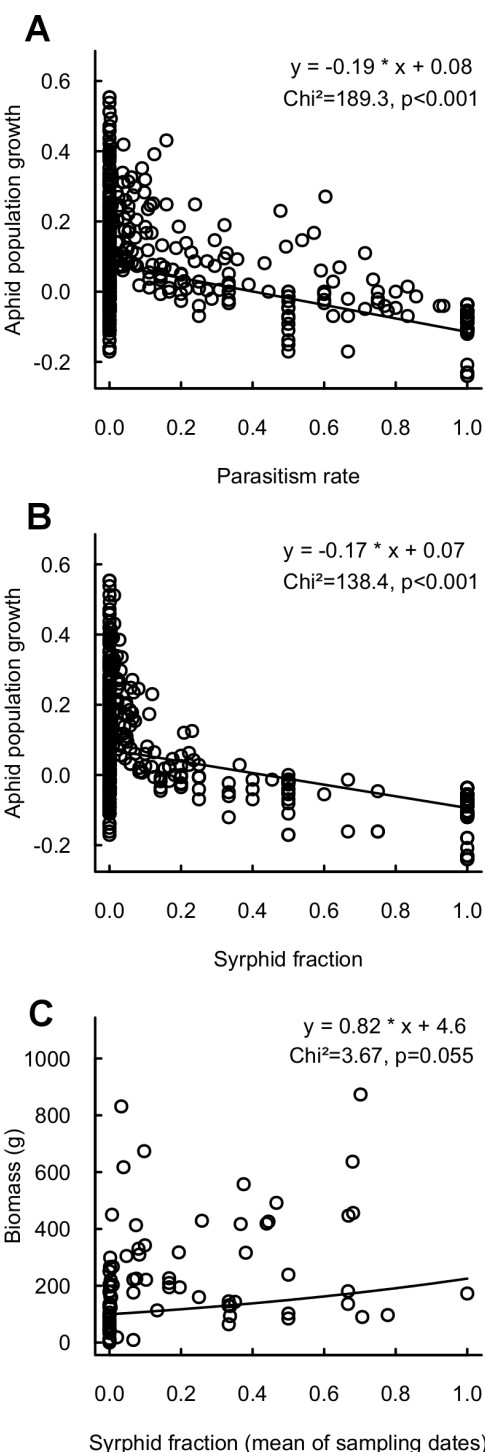

**Figure 3** Relationship between aphid population growth and (A) parasitism rate (*n* = 601), (B) syrphid fraction (*n* = 588) and (C) between final cabbage biomass and syrphid fraction (mean of sampling dates; *n* = 84).

## DISCUSSION

### Pest control across the landscape gradient

This study shows that in the absence of pressure by natural enemies, aphid population growth was higher in complex than in simple landscapes. However, aphids were also strongly reduced by natural enemies, and this pest reduction increased with landscape complexity. Overall, aphids were thus maintained at low levels in all landscapes, because higher aphid growth and colonization in complex landscapes was compensated by stronger pest control by natural enemies.

An increase of aphid population growth, but also of aphid suppression with landscape complexity, can be explained by higher availability of alternative resources and overwintering habitats in seminatural areas around fields, which may benefit colonization and reproduction in fields by both pests and their natural enemies in the course of the growing season (*Thies, Roschewitz & Tscharntke, 2005*; *Bianchi, Booij & Tscharntke, 2006*; *Holzschuh, Steffan-Dewenter & Tscharntke, 2010*). Here, stronger aphid population growth was found in complex landscapes in treatments excluding natural enemies, where aphids were enclosed in fine mesh cages as a barrier to natural enemies. Such higher growth rates may have been caused by an additional factor such as differences in the nitrogen content of fields (*Butler, Garratt & Leather, 2012*). In this study, known management differences did not correlate with landscape context, but interactions of these factors could not be tested due to a lower range of landscape values surrounding conventional fields. In addition, cages were permeable to some extent to colonization by aphids (see Methods). Although densities in plots at the start of the experiment were not predicted by the landscape context, aphid colonization is likely to have taken place in the course of the experiment, particularly in the first phase of the growing season (until sampling date 1). Thus, a positive impact of seminatural habitats on aphid colonization is the most plausible explanation for higher aphid growth rates in complex landscapes in the absence of enemies.

Several studies have measured the distribution of pest abundance across landscapes, but results remain inconclusive overall, as pests appear to decrease, increase or not to vary with landscape complexity (*Bianchi, Booij & Tscharntke, 2006*; *Chaplin-Kramer et al., 2011*). However, only a few recent studies have measured landscape effects on pests in the absence vs. in the presence of natural enemies and thus provide measures of actual pest pressure across landscapes (*Gardiner et al., 2009*; *Thies et al., 2011*; *Chaplin-Kramer & Kremen, 2012*; *Holland et al., 2012*; *Martin et al., 2013*; *Rusch et al., 2013*). Of these, three report results of pest variation separately from an aggregated measure of pest control, with pest pressure either increasing with landscape complexity as here (*Chaplin-Kramer & Kremen, 2012*; *Martin et al., 2013*) or decreasing with combined landscape and local extensification (*Thies et al., 2011*). The lack of general patterns highlights the fact that pest densities are affected by the landscape both directly and indirectly through landscape effects on higher trophic levels (enemies) and emphasizes the need to experimentally address these factors in isolation from each other.

In agreement with our results, the few studies measuring actual pest control of aphids as the difference between pest density in the presence and in the absence of natural enemies,

also find increasing intensity of pest control with the proportion of natural or seminatural habitats in the surrounding landscape. The strength of aphid pest control increased in these studies by a factor of two to five from simple to complex landscapes (*Gardiner et al., 2009*; *Chaplin-Kramer & Kremen, 2012*; *Rusch et al., 2013*), and was here on average six times higher in complex than in simple landscapes. So far, however, no other study has disentangled the single and combined contributions of antagonist guilds including birds to aphid pest control along a landscape complexity gradient.

## Enemy contributions to pest control and interactions

On average, our results suggest that flying insect enemies had stronger impacts than ground-dwellers on aphid control. Further, these effects appeared to increase in complex landscapes, paralleling the generally higher abundance and species richness of these enemies in landscapes with high amounts of (well-connected) seminatural habitat, than in simple ones where overwintering, nesting, and food resources are rare (*Bianchi, Booij & Tscharntke, 2006*; *Holzschuh, Steffan-Dewenter & Tscharntke, 2010*; *Chaplin-Kramer et al., 2011*). The importance of landscape complexity for flying insect effectiveness is confirmed by increased activity rates (parasitism and syrphid fractions) of this guild in complex landscapes, both here and in other studies (e.g., *Thies & Tscharntke, 1999*; *Rand, Van Veen & Tscharntke, 2012*). However, increased parasitism of aphids is likely to be accompanied by even stronger hyperparasitism in complex landscapes, because sensitivity of these organisms to the landscape context has been shown to increase with trophic level (*Rand, Van Veen & Tscharntke, 2012*). The parasitism rates observed here thus likely reflect the outcome of interactions with the 4th trophic level in the previous year, and indicate that parasitism was maintained in complex landscapes despite possible pressure by hyperparasitoids.

Control by ground-dwellers increased to a lesser extent with landscape complexity than control by flying insect enemies. As a result, the relative contribution of these guilds to pest suppression was influenced by the landscape context. Stronger effects of flying insects compared to ground-dwellers in complex landscapes are in agreement with previous studies (*Schmidt et al., 2003*; *Thies et al., 2011*; but see *Safarzoda et al., 2014*) and with the idea that generalist predators (ground-dwellers) have lower impacts than specialists when prey density is high (*Straub, Finke & Snyder, 2008*). Indeed, within our study system, relative prey densities were highest in these landscapes. Birds, in contrast, showed no clear contribution to reducing aphids. Thus, in addition to generally showing predictable responses to changes in landscape complexity (*Bianchi, Booij & Tscharntke, 2006*), the guild of flying insects had the strongest impact on aphids under conditions of high landscape complexity.

Strong impacts of flying insects are confirmed by the negative relationships between aphid population growth rates and parasitism and syrphid fractions. However, of these, only syrphids had a positive impact on crop biomass, the final measure of interest for assessment of pest control (in this experiment, landscape effects on biomass were linked to the Lepidopteran pest complex; *Martin et al., 2013*). Ultimately, benefits for farmers may be higher when enemies are predators that immediately suppress pests, than when

they are parasitoids with slower impacts on their hosts. Overall, flying insect enemies and particularly syrphid flies may thus represent an optimal focus for efforts to maximize natural pest control in agricultural landscapes. However, the effectiveness of these efforts depends on the balance between individual enemy contributions, and how they interact with other natural enemy guilds.

Interactions between flying insects and ground-dwellers led to complementary effects on pest control, as aphid suppression was stronger in the presence of both guilds than with either guild alone, in agreement with results in other aphid systems (*Safarzoda et al., 2014*; *Schmidt et al., 2003*). This effect may be due to density-dependent predation by each guild, to their spatially segregated foraging (*Straub, Finke & Snyder, 2008*), but also to escape behavior of aphids from flying insects increasing the chances of ground-dweller predation (*Losey & Denno, 1998*). In this system, however, escape behavior of the aphids was neither observed in the field (EA Martin, pers. obs., 2010) nor appears to be documented for the species considered; in contrast, several species of ground-dwellers present in this region (spiders, carabids and staphylinids) are known to forage by climbing directly onto crop plants (*Hannam, Liebherr & Hajek, 2008*; *Suenaga & Hamamura, 1998*). In the absence of antagonist interactions, access to both guilds may increase overall enemy density and thus benefit aphid control. Syrphid fractions show that syrphid larvae were little influenced by the presence of ground-dwellers and could thus suppress aphids independently of ground-dweller activity. Lower parasitism in treatments accessible to ground-dwellers indicate that in complex landscapes, ground-dwellers may have preyed not only on live aphids, but also or preferentially on parasitized mummies, as observed in local-scale studies (*Snyder & Ives, 2001*). However, this did not hinder the overall complementarity of these guilds for aphid suppression.

Birds, the largest and most generalist predators in the system, appeared to interact in complex ways with aphids and other enemies. Although their effects have rarely been quantified in agricultural systems with annual crops (*Mooney et al., 2010*) and almost never in the light of interactions with other enemies (but see *Hooks, Pandey & Johnson, 2003*; *Martin et al., 2013*), birds are known to occasionally feed on aphids in these systems (*Tremblay, Mineau & Stewart, 2001*). Here, predation by birds of parasitized aphids and syrphids in complex landscapes is suggested by lower enemy rates in bird-accessible treatments, at least while aphid populations are high (sampling dates 1 & 2). Bird predation on parasitoids vs. syrphids has different consequences: coincidental predation of mummies may still decrease aphid densities, but omnivorous predation of syrphids should theoretically release the shared prey (*Straub, Finke & Snyder, 2008*). Overall, effects on aphids may thus level out and, as found here, lead to no clear effect of birds on aphid population growth.

Overall, strong negative effects of intraguild predation on aphid suppression were not found, as aphids were reduced sufficiently by the combination of all guilds to remain at low levels in open treatments throughout the experiment. This result is in contrast to the disruptive effects of intraguild predation by birds found for Lepidopteran pest control in the same system (*Martin et al., 2013*). It thus emphasizes that effects of natural

enemies and particularly birds are pest organism-dependent. In the case of aphids, pest control provided by the combination of three enemy guilds was higher than pest control by individual guilds, and this result held true across landscapes with increasing complexity. These results support the idea that higher functional diversity may benefit ecosystem function and services across large spatial scales (*Cardinale et al., 2006*), for particular combinations of functions and guilds. However, consideration of additional guilds or different pests may greatly influence this relationship, as suggested by contrasting effects of enemy functional diversity on control of Lepidopteran pests in the same experiment (*Martin et al., 2013*).

Interactions among pests, particularly between Lepidoptera and aphids, may also take place that influence pest population growth, predation, parasitism and enemy-level interactions. Though not testable by the present design, among-pest interactions merit further investigation, particularly their potential response to variations in enemy density and community composition.

### Natural enemies and pests in organic vs. conventional plots

In plots surrounded by conventional fields, soil nitrogen availability was higher than near organic fields (*Martin et al., 2013*) and is likely to be responsible for higher initial aphid population build-up in these in plots (*Butler, Garratt & Leather, 2012*). More natural enemies than initially present were thus required near conventional fields to effectively constrain aphids to the same degree as near organic fields. Higher population growth at sampling date 1 in conventional fields, followed by no differences at subsequent dates, suggest that a time lag took place near conventional fields before enemies reached sufficient densities to effectively reduce pests in these plots (*Krauss, Gallenberger & Steffan-Dewenter, 2011*). This supports the idea that early-stage pest control was less efficient near conventional compared to organic fields. In contrast, strong pest regulation near conventional fields later in the season indicates that enemies responded with strong positive density-dependence to the initial population build-up of aphids in these fields, which may have been caused by the emission of enemy-recruiting volatiles by the plants under herbivore attack (*Kessler & Baldwin, 2001*; *Thaler, 1999*).

## CONCLUSION

Despite complex interactions occurring between enemy functional guilds across landscapes, pest control of aphids benefited in all landscapes from high enemy functional diversity, and was stronger in complex landscapes with high amounts of seminatural habitat than in simple ones. Aphid pest control by flying insects and ground-dwellers was complementary, but flying insects including syrphids provided the strongest contributions to aphid pest control particularly in complex landscapes. To our knowledge, this study is the first to provide results of natural enemy interactions for aphid pest control outside of Europe and the USA. These results emphasize the need to identify underlying interaction mechanisms of pest control at large spatial scales, in order to provide realistic predictions of ecosystem service provision in agricultural landscapes worldwide, and thus improve the applicability of this concept for higher agricultural sustainability.

## ACKNOWLEDGEMENTS

We thank the farmers of Haean and M Ahn for permission to use their fields. J Bae provided translation and logistic help. M Hoffmeister, G-H Im, P Poppenborg and S Lindner provided field assistance.

### Funding

This study was funded by the Deutsche Forschungsgemeinschaft (DFG) in the Bayreuth Center of Ecology and Environmental Research BayCEER international research training group TERRECO: Complex Terrain and Ecological Heterogeneity. The funders had no role in study design, data collection and analysis, decision to publish, or preparation of the manuscript.

### Grant Disclosures

The following grant information was disclosed by the authors:
Deutsche Forschungsgemeinschaft (DFG).

### Competing Interests

The authors declare there are no competing interests.

### Author Contributions

- Emily A. Martin conceived and designed the experiments, performed the experiments, analyzed the data, wrote the paper, prepared figures and/or tables, reviewed drafts of the paper.
- Björn Reineking reviewed drafts of the paper and discussed data analysis.
- Bumsuk Seo performed landscape mapping.
- Ingolf Steffan-Dewenter conceived and designed the experiments, discussed data analysis and reviewed drafts of the paper.

### Data Deposition

The following information was supplied regarding the deposition of related data:
Dryad repository: DOI 10.5061/dryad.n6428.

### Supplemental Information

Supplemental information for this article can be found online at http://dx.doi.org/10.7717/peerj.1095#supplemental-information.

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
