# Peer review of "Pest control of aphids depends on landscape complexity and natural enemy interactions"

_PeerJ, doi:10.7717/peerj.1095_

## Round 0.1 · original submission · Minor Revisions

Both referees suggest minor revisions, and I agree. This is a well-written MS, and the data interpreted reasonably and are nicely presented. Both referees have provided very helpful and expert reviews, and I look forward to a revised MS from the authors, along with a detailed reply letter.

·

Basic reporting

No comments

Experimental design

Very nicely done.

Validity of the findings

Very nicely done.

Additional comments

This paper examines the effect of landscape complexity on the growth of aphid populations exposed to varied functional guilds of predators. This study is very well-written, interesting and competently executed. I have very few points to raise, and they are primarily points for discussion, rather than must-edits, and a few minor points of clarity.

My main point for clarification is in the results section. Starting at line 324, the authors describe results “aphid population growth was four times lower…” which, although it’s correct, is a bit non-intuitive- as a reader, when I was trying to interpret this and the following similar statements, I had to read it several times. “Growth” and ‘times” both imply a quantity is getting bigger, but the sentence describes a reduction- it has the feeling of a double negative. I think the results could be worded more directly, either by switching the order of the comparison (“LargeResult was N times larger than SmallResult”) or by removing the implication of multiplication (“SmallResult only reached one tenth the value of LargeResult”). I’d prefer to see either of these approaches used throughout the results section.

I also feel the use of ‘round’ for sample date is a bit ambiguous. I suggest using the actual sample date, or calling the variable ‘sample date’ instead.

Line 96- a colleague of mine led a few interesting studies in this area in field crops and grasslands, and linked it to insecticide use:
Werling, B. P., T. D. Meehan, B. A. Robertson, C. Gratton, and D. A. Landis. 2011. Biocontrol potential varies with changes in biofuel–crop plant communities and landscape perenniality. GCB Bioenergy 3:347-359.
Meehan, T. D., B. P. Werling, D. A. Landis, and C. Gratton. 2011. Agricultural landscape simplification and insecticide use in the Midwestern United States. Proceedings of the National Academy of Sciences 108:11500-11505.

Line 135- Were coccinellids important in this system? I ask only because they’re so dominant in North American agricultural systems!

174-176- I’m confused by this sentence. I can glean the meaning after I read the rest of the paragraph, but consider rewording this first sentence for clarity.

239- What does n refer to here? The number of ‘growth’ datapoints? Total number of aphids?

441-450- I recently worked with a student who examined the effect of foliar versus ground dwelling predators in early season cereal aphids, and she found a net greater impact of ground-dwelling predators- although she didn’t specifically quantify landscape context, it would be interesting to go back and see where her work would fit in the context of these models- her study is here:
Safarzoda, S., C. A. Bahlai, A. F. Fox, and D. A. Landis. 2014. The role of natural enemy foraging guilds in controlling cereal aphids in Michigan wheat. PLOS One:10.1371/journal.pone.0114230.

469-471- Just out of curiosity, was the community of ground-dwellers assessed for members that can/do climb plants? Any chance that there was an interaction with foliar foragers- ie foliar predators causing dropping behaviors (do cabbage aphids drop when disturbed?)

·

Basic reporting

The article is clearly written, places the research into context well, and was easy to understand and follow. I appreciated the clarity of the figures and the icons to show the different exclusion treatments. I have some minor comments below that might help with manuscript clarity:

L35 suggest changing the wording to “but was reduced by natural enemies to similar growth rates across all landscapes.”
L37: suggest changing “was formed by” to “supplied by” or similar
L92-94: suggest changing to “estimating the effects of landscape context on pest suppression requires distinguishing pest abundance both in the presence and absence of enemies.”
L107: suggest changing to “interactions of predator species and…”
L116: suggest changing to “Quantifying effective pest control across alndscapes thus requires…”
L124-125: move “(“flying insects”) to earlier in the sentence, at the moment it appears that this refers to the aphids themselves instead of flying aphid predators.
L141: change to “We hypothesized…”
L143: “spillover” hasn’t been defined or mentioned before but probably should be. Could also refer to Tscharntke et al. 2005. Ann. Zool. Fennici 42: 421-432 for this.
L397: suggest changing wording to “Here, stronger aphid population growth…”
L398: I’m unclear what is meant by “aphids were relatively isolated from their surroundings” Do you mean isolated from predators?
L437-439: I’m not familiar with the terms “primary parasitism’ and ‘secondary parasitism’ and these haven’t been mentioned earlier. Is this referring to parasitism and hyperparasitism? Might need a bit more explanation to place in context.

Experimental design

In general, the research questions and methods were clearly described and the study is of high quality. I do have a few questions with regards to the statistical analyses below that should be addressed before publication:

L162: Why was 700m used as the radius around fields to determine landscape complexity? This is likely a reasonable distance for this system, but no justification for this is given.
L250-252: Why would less colonization of cabbage plants in the fine mesh cage treatment lead to an underestimation of pest control? This might need just a bit more explanation. Could this effect also lead to an overestimation of the level of control provided by ground beetles?
L260: What does the psi symbol here refer to?
L265-269: Sampling round is included in the analysis as an explanatory variable, but these different time periods aren’t independent. I would think that this variable should be treated as a repeated measure or that sampling round should be included as a nested random effect to account for this non-independence. This might not in the end change the overall results, but this non-independence should be addressed.
L282-283: Did including the initial aphid densities in the models affect the model-averaged coefficients for the explanatory variables? It seems that the models including initial aphid densities almost always have lower AICc values than the models without, so these models seem to be better supported than the ones presented in the main text. Thus, it's not entirely clear to me why the models without this variable are instead presented in the main text.
L296-297: Are the "predicted values" referred to here the coefficients of the explanatory variables from the model averaging?

Validity of the findings

I found the findings generally well presented and rationally sound. The overall conclusions of the study are well supported. It's not clear if the data in the manuscript have been deposited in a discipline-specific repository, but I assume this would happen before publication. Some minor comments with respect to the Results & Discussion sections:

L290: Should Table S2 present the results for parasitism rates and syrphid fractions as well?
L319-321: Are these densities of aphids large enough to potentially impact cabbage growth or biomass? I’ve worked with soybean plants and aphids and they need to have aphid densities at least >250 aphids/plant before biomass is affected. Is it similar for cabbage?
L333-337: This effect that is mentioned seems primarily to be present in Round 1, and weakens over time to Round 3 (Fig. 1). This should probably be mentioned here.
L345: not sure how Table S3 shows this result. It appears that the slope coefficients in the table are averaged over all of the different landscape complexities. Same for L341 and L348.
L362-363: One of the results that stands out here to me is the different trend for –F-G and –F-B with parasitism rate in Round 3 in Figure 2. This doesn’t seem to be mentioned, is there a reason this was ignored or isn’t interesting?
L446-447: were the overall levels of aphids observed in this study high or low? This could affect the results you saw and the generality of the conclusions you reach.
L486-487: It’s not clear to me why the result mentioned emphasizes that effects of natural enemies and birds are pest organisms-dependent. Is the point that the effects of enemy functional diversity are different between aphids and Lepidopteran pests? If so, this should be more obvious to readers not familiar with Martin et al. 2013.
L500: I found it somewhat counter-intuitive that conventional fields appeared to have greater pest regulation than organic. While not the main focus of the study, I would have like to have seen a bit more discussion of this result and comparison to other studies.

Additional comments

This paper addresses the question of how agricultural landscape complexity impacts pest regulation of aphids, and more specifically, how this impacts interactions between different natural enemy guilds and the subsequent strength of pest regulation. An experimental approach is used where different enemy guilds are excluded from cabbage plants across a gradient of landscape complexity is used to tease apart these interactions and impacts. The authors find that both aphid population growth and pest regulation increase with landscape complexity, with complementary interactions between predator guilds.

In general, I enjoyed reading this manuscript and found it well written and clear, with clearly presented results and discussion around these results, despite the complexity of the experimental design. The authors were able to advance knowledge about how landscape structure simultaneously affects pest pressure, pest predation, and crop yield, all of which are important components of pest regulation. I include some minor comments and suggestions below to help improve the clarity of the manuscript, particularly around the description and appropriateness of the study’s methods and statistical analysis.

---

## Round 0.2 · accepted · Accept

Both reviewers recommended minor revisions, and the authors have completed a very thorough, detailed, and satisfactory response to the reviewers' questions, comments, and concerns. The resulting revised MS represents an improvement, although as noted by the reviewers the initial submission was quite good already.

I'd like to thank the reviewers for their careful reading and commenting, and the authors for their exemplary response. As both reviewers have allowed their names to stand as public reviews, I'd encourage the authors to also release the results of the review as a public document to stand alongside their published MS.